# Present and Future of Parkinson’s Disease in Spain: PARKINSON-2030 Delphi Project

**DOI:** 10.3390/brainsci11081027

**Published:** 2021-07-31

**Authors:** Diego Santos García, Marta Blázquez-Estrada, Matilde Calopa, Francisco Escamilla-Sevilla, Eric Freire, Pedro J. García Ruiz, Francisco Grandas, Jaime Kulisevsky, Lydia López-Manzanares, Juan Carlos Martínez Castrillo, Pablo Mir, Javier Pagonabarraga, Francisco Pérez-Errazquin, José María Salom, Beatriz Tijero, Francesc Valldeoriola, Rosa Yáñez, Arantxa Avilés, María-Rosario Luquín

**Affiliations:** 1Department of Neurology, Complexo Universitario de A Coruña (CHUAC), C/As Xubias 84, 15006 A Coruña, Spain; 2Department of Neurology, Hospital Universitario Central de Asturias, Avenida de Roma s/n, 33011 Oviedo, Spain; marta.blazquez.estrada@gmail.com; 3Neurology Service, Hospital Universitari de Bellvitge, 08907 Barcelona, Spain; mcalopa@bellvitgehospital.cat; 4Unidad de Trastornos del Movimiento, Servicio de Neurología, Hospital Universitario Virgen de las Nieves, Instituto de Investigación Biosanitaria ibs. Granada, 18013 Granada, Spain; fescamilla@hotmail.com; 5Departamento de Neurología, Hospital IMED Elche, Calle Max Planck 3, 03203 Elche, Spain; dr.freyre@gmail.com; 6Servicio de Neurología, Fundación Jiménez Díaz, Avda Reyes Católicos 2, 28040 Madrid, Spain; PGarcia@fjd.es; 7Servicio de Neurología, Hospital Gregorio Marañón, Calle del Dr. Esquerdo 46, 28007 Madrid, Spain; francisco.grandas@salud.madrid.org; 8Servicio de Neurología, Unidad de Trastornos del Movimiento, Hospital de la Santa Creu i Sant Pau, Universitat Autònoma de Barcelona, CIBERNED, C/Mas Casanovas 90, 08041 Barcelona, Spain; jkulisevsky@santpau.cat (J.K.); pmir@us.es (P.M.); JPagonabarraga@santpau.cat (J.P.); 9Servicio de Neurología, Hospital de la Princesa, Calle de Diego de León 62, 28006 Madrid, Spain; lydia.lopez@salud.madrid.org; 10Servicio de Neurología, IRYCIS, Hospital Universitario Ramón y Cajal, M-607, km. 9, 100, 28034 Madrid, Spain; jcmcastrillo@gmail.com; 11Unidad de Trastornos del Movimiento, Servicio de Neurología y Neurofisiología Clínica, Instituto de Biomedicina de Sevilla, Hospital Universitario Virgen del Rocío, Av. Manuel Siurot, S/n, 41013 Sevilla, Spain; 12Servicio de Neurología, Hospital Virgen de la Victoria, Campus de Teatinos, S/N, 29010 Málaga, Spain; pacoerrazquin@hotmail.com; 13Servicio de Neurología, Hospital Clínico Universitario de Valencia, Avda Blasco Ibañez No. 17, 46010 Valencia, Spain; jomasaju@hotmail.es; 14Servicio de Neurología, Hospital Cruces, Cruces Plaza, S/N, 48903 Barakaldo, Bilbao, Spain; beatriz_tijero@hotmail.com; 15Parkinson’s Disease and Movement Disorders Unit—Neurology Service, Hospital Clinic Barcelona, Universitat de Barcelona & Institut d’Investigacions Biomèdiques August Pi i Sunyer (IDIBAPS), C/Casanova 170, 08036 Barcelona, Spain; fvallde@clinic.cat; 16CIBERNED, C/Mas Casanova 170, 08041 Barcelona, Spain; 17Servicio de Neurología, Complejo Hospitalario Universitario de Ourense, Ramon Puga Noguerol 54, 32005 Ourense, Spain; ryanezb@telefonica.net; 18Departamento Médico, Zambon S.A.U. C/Maresme, 5 Pol. Ind. Can Bernades-Subirà, 08130 Barcelona, Spain; Arantxa.Aviles@zambongroup.com; 19Departamento de Neurología, Clínica Universidad de Navarra, Instituto de Investigación Sanitaria de Navarra, Avenida de Pio XII 36, 31008 Pamplona, Spain; rluquin@unav.es

**Keywords:** diagnosis, economic impact, epidemiology, management, mortality, Parkinson’s disease, quality of life, Spain, treatment

## Abstract

Parkinson’s disease (PD) is a chronic progressive and irreversible disease and the second most common neurodegenerative disease worldwide. In Spain, it affects around 120.000–150.000 individuals, and its prevalence is estimated to increase in the future. PD has a great impact on patients’ and caregivers’ lives and also entails a substantial socioeconomic burden. The aim of the present study was to examine the current situation and the 10-year PD forecast for Spain in order to optimize and design future management strategies. This study was performed using the modified Delphi method to try to obtain a consensus among a panel of movement disorders experts. According to the panel, future PD management will improve diagnostic capacity and follow-up, it will include multidisciplinary teams, and innovative treatments will be developed. The expansion of new technologies and studies on biomarkers will have an impact on future PD management, leading to more accurate diagnoses, prognoses, and individualized therapies. However, the socio-economic impact of the disease will continue to be significant by 2030, especially for patients in advanced stages. This study highlighted the unmet needs in diagnosis and treatment and how crucial it is to establish recommendations for future diagnostic and therapeutic management of PD.

## 1. Introduction

Parkinson’s disease (PD) is the second most common neurodegenerative disease after Alzheimer’s disease [1] and affects around 120.000–150.000 individuals in Spain [2]. Excellent symptomatic treatment is available, although no neuroprotective medication has been found so far. Even though the average age of onset is 60 years, the number of diagnoses under the age of 50 is increasing [3]. Given that PD prevalence increases with age [3], and considering the increase of life expectancy, the number of people affected by PD is predicted to strongly increase in the future [4].

PD is characterized by motor symptoms including bradykinesia, resting tremor and muscle rigidity, but also by non-motor symptoms. These include cognitive impairment, mood disorders and depression, and may appear even at early disease stages [5,6]. Moreover, non-motor symptoms are often misdiagnosed because PD is mainly diagnosed on clinical motor symptomatology, following the United Kingdom Parkinson’s Disease Society Brain Bank (UK-PDSBB) criteria [7,8]. The Movement Disorder Society’s clinical diagnostic criteria for PD (MDS-PD) are intended for use specifically in research; however, they can be used as a general guide for clinical diagnosis. MDS-PD include supportive criteria apart from motor symptoms [9]. Non-motor symptoms in the prodromal phase are usually not very specific (except for rapid eye movement [REM] sleep behavior disorder [RBD]), so they are not strong enough by themselves to reach a PD diagnosis. Recent studies have allowed PD diagnosis at prodromal stages in individuals with risk factors (hyposmia, depression and constipation, among others) or RBD itself [10]. PD’s heterogeneous presentation reflects its complex and multifactorial etiogenesis, its diverse clinical course [11] and probably a different pathogenic mechanism. Therefore, PD management is a major challenge because clinical response to treatment is also heterogeneous and disease treatment has to be personalized.

There is no first-choice treatment for PD management, although many pharmacological and non-pharmacological strategies are available. Among pharmacological treatments, levodopa, dopaminergic agonists (DA), catechol-O-methyltransferase inhibitors (COMTI) and monoaminooxidase B inhibitors (MAOI-B) are the most common therapies for PD, used both alone or in combination [3]. Since several motor, non-motor symptoms and related complications must be treated, PD management is often complex.

As a chronic progressive disease, PD has a great impact on patients’ and caregivers’ private, social and professional lives. PD caused more than 200,000 deaths and 3.2 million disability-adjusted life-years (DALYs) worldwide in 2016, and more than 4000 deaths and 54,175 DALYs in Spain [12]. Moreover, it also causes substantial societal burden and entails an important economic impact [13].

Considering the high incidence of the disease, its increased prevalence due to population aging and the unmet needs regarding diagnosis and treatment, it is crucial to better understand PD management to optimize and design future strategies. Thus, the aim of the PARKINSON 2030 project was to discuss and reach a consensus using the Delphi method (a process used to arrive at a group opinion or decision by surveying a panel of experts in which the experts respond to several rounds of questionnaires, and the responses are aggregated and shared with the group after each round) among a panel of experts in movement disorders on the current situation, their 10-year forecast of the general management of PD in daily clinical practice and to establish recommendations on the diagnostic and therapeutic management of PD.

## 2. Materials and Methods

We specifically used a modified Delphi method (a group consensus strategy that systematically uses literature reviews, opinions of stakeholders and the judgment of experts within a field to reach agreement) [14,15,16] to better understand the management and the socioeconomic burden of PD in Spain. This Delphi approach was carried out in 6 successive phases: creation of an Advisory Committee, definition of criteria to select panelists, design of the Delphi questionnaire, Delphi survey administration, data collection and data analysis.

The Advisory Committee included 4 experts in movement disorders, members of the Study Group of Movement Disorders of the Spanish Neurology Society (Sociedad Española de Neurología, SEN). The Advisory Committee defined the project and created the first version of the Delphi questionnaire.

Panelists included were experts in movement disorders from different regions of Spain. The list of participating panelists is shown in Appendix A.

### Preparation and Administration of the Questionnaire and Data Analysis

A literature search that included guidelines, reviews and other types of critical synthesis of scientific literature, as well as a bottom-up search from the main references to identify other papers of interest was carried out, and the initial questionnaire developed. The questionnaire addressed nine topics: the epidemiology of PD, the pathophysiology of PD, PD’s impact on patients and caregivers, PD diagnosis, PD follow-up, PD treatment, the economic impact of PD, the current and future situation of PD and the role of patient and caregiver associations. The initial version of the questionnaire was validated by 14 additional experts in movement disorders, acting as the National Committee of experts (Appendix A). The questionnaire was held to two rounds of voting between October–December 2019 (first wave) and September–November 2020 (second wave). Panelists completed the questionnaire through an online platform that ensured data anonymity and confidentiality.

All questions about the current situation used a 4-level scale, which varied depending on the question (never, rarely, often, always; or, strongly disagree, disagree, agree, strongly agree; or, not important, not very important, important, very important; or, not useful, not very useful, useful, very useful; or, no impact, little impact, quite some impact, a lot of impact). All questions referring to future trends used a nine-point Likert-type ordinal scale, divided in three categories: 1 to 3 (it will decrease), 4 to 6 (it will not change) and 7 to 9 (it will increase). When ≥70% of panelists agreed in one category of response, that item was considered to have reached “consensus” on that category. Otherwise, the item was stated as “undetermined” and sent for a second round of votes, together with all comments made by panelists in face-to-face meetings.

Between the two rounds of voting, panelists attended 7 face-to-face local meetings moderated by the members of the National Committee of experts. In these meetings, the results of wave 1 were presented and discussed, and the items of the questionnaire to include in wave 2 were agreed upon. Although 12 meetings were planned, due to the state of alarm decree by COVID-19 in March 2020, only 7 meetings were held. All items that did not reach consensus in the first wave were included in the second wave.

The final results of the Delphi survey were further evaluated and discussed by the Advisory and National Committees.

## 3. Results

### 3.1. Panelists’ Description

Seventy-five (75) experts in movement disorders from different regions of Spain were included in the panel, and a mean of 14.3 years of experience was reported. Regarding the type of work center, 74% had practiced in tertiary centers, while only 4% worked in primary care centers. A full description of the participant panelists can be found in the Appendix A.

### 3.2. Epidemiology of PD—Etiopathogenesis

Overall, 92% and 84% of the experts consulted considered that prevalence and incidence of PD will increase by 2030, respectively. The trend towards 2030 in the percentage of PD patients diagnosed at <50 years was not agreed upon by the experts: 67% indicated that this percentage will not change and 33% indicated that it will increase. Regarding the etiopathogenesis of the disease, 73% of experts considered that the identification of non-modifiable PD risk factors will not progress in 2030, while knowledge of modifiable PD risk factors will not be improved (76%) (data not shown).

### 3.3. Prognostic Factors

According to the experts, the factors that are currently widely or always used as prognostic factors for PD include phenotype with axial involvement (100%), cognitive impairment (99%), early presence and severity of motor complications (99%), age at onset (93%), neuropsychiatric disorders (92%) and Hoehn & Yahr clinical stage (85%). Experts reached a consensus that, by 2030, only the assessment of neuropsychiatric disorders as a PD prognostic factor will increase. Among the prognostic factors less used nowadays, the experts affirmed that the use of genetic testing, structural imaging, CSF (cerebrospinal fluid) biomarkers and serum molecular markers will increase in the future (Table 1).

### 3.4. Medical Specialties Involved in the Management of the PD Patient

Nowadays, a poorly multidisciplinary approach is applied in the treatment of PD and it is generally limited to the neurologists specialized in PD and general neurologists, with no significant involvement of other disciplines. However, panelists agreed that the role of movement disorder specialists in nursing would gain importance especially in the treatment of PD patients with advanced stages and in patients under advanced therapy (Figure 1).

### 3.5. Diagnosis

Overall, 83% of the experts agreed with the definition of the diagnosis of the MDS criteria [7,8]; however, there was no consensus on the future trend (Table 2). Among the diagnostic tools currently applied, only the use of functional neuroimaging (78%) was predicted to increase by 2030. Atypical signs and non-motor symptoms are also currently considered key aspects for PD diagnosis, by 95% and 79% of panelists, respectively, without consensus on future trends. Regarding other biomarkers experts suggest that only the use of biochemical markers (74%) and genetic criteria (70%) will increase by 2030 (Table 2).

By 2030, 94% of panelists considered that the time taken to obtain a diagnosis will decrease compared to today, and 84% agreed that the percentage of misdiagnosed patients will decrease (data not shown).

### 3.6. Follow-Up

Concerning the tools used in the clinical follow-up, the Hoehn & Yahr scale (88%) and the scale of the Movement Disorders Society—Unified Parkinson’s disease rating scale (MDS-UPDRS) (77%) are predicted to be used in the same frequency in the future. In contrast, panelists agreed that several tools not widely used nowadays will be progressively implemented by 2030. These include wearable, non-wearable, hybrid and smartphone sensors (94%), pharmacogenetics (72%) and biochemical markers (71%) (Table 3). There is currently no consensus on the contribution of new technologies to the remote monitoring of PD patients, however, panelists agreed that their contribution will increase by 2030 (Appendix A).

### 3.7. Treatment

According to the experts, the use of the currently most frequent monotherapy treatments for non-fluctuating patients (MAOI, DA and levodopa < 600 mg) will not change by 2030. Although it did not reach consensus, panelists observed a trend to decrease in the use of levodopa at low–moderate doses (<600 mg) (35%) and anticholinergics in monotherapy (35%) by 2030. In addition, a trend to increase in the use of safinamide in monotherapy (56%) was observed, while no increase for other MAOI agents in monotherapy was predicted (Table 4). Although experts did not agree, a trend to increase the combination of levodopa + safinamide (from 8.7% to 63%) and DA + safinamide (from 4.2% to 51%) was observed, which was not observed for other MAOI agents in combination (6% in the future for both combinations). The use of three-drug combination will also remain unchanged in the future. Finally, a trend to increase the combination levodopa + safinamide + DA (from 6.9% to 59%) was observed.

Up to 3.9% of non-fluctuating patients are currently treated with polytherapy (≥4 drugs) with no predicted changes in the future.

Overall, treatment of fluctuating patients remains unchanged in the future (Table 4). Although it did not reach consensus, we observed a trend to increase in the use of safinamide in combination with levodopa (63% in the future) or in combination with DA + levodopa (59% in the future), not observed when other MAOI agents are considered in combination with levodopa (3%) or in combination with DA + levodopa (8%) (Table 4). Only 20% of the experts consider that the combination COMTI + DA + levodopa will increase. As expected, polytherapy is frequent in the treatment of fluctuating patients and will remain unchanged in the future (77%) (Table 4).

Regarding patients in advanced therapy, deep brain stimulation (DBS) treatment was the only treatment predicted to increase (76%). Although without consensus, a trend towards an increased use of apomorphine s.c. or carbidopa/levodopa (enteral) in combination with other pharmacological treatments (52% and 68%, respectively) was observed (Table 4).

Just as important as the introduction of drugs in the management of PD patients is the withdrawal of pharmacological treatments in patients already in palliative care. There is consensus that the first drugs to be withdrawn at present are anticholinergics (93%) and amantadine (80%) and similar rates of agreement for the future (92% and 81%, respectively) were observed. The last drug to be withdrawn is levodopa, either nowadays (91%) or in the future (89%) (data not shown).

With regard to symptomatology and/or comorbidities as a reason for changing treatment, the experts showed a high degree of agreement that the following are reasons for changing PD treatment: onset of neuropsychiatric and cognitive disorders; deterioration in basic activities of daily living (BADL) autonomy; motor symptoms; impaired quality of life; affective, emotional and volitional disorders; sleep disorders; and autonomic disorders. In contrast, disorders of other organs outside the central nervous system (CNS) (osteoporosis, back pain, respiratory disorders) would not be a reason to reconsider treatment, according to 84% of the experts (Table 5).

The experts showed a high degree of agreement in relation to the statements concerning the change of treatment for PD, except for the item neuroprotective treatments prevail over symptomatic treatments, which reached a high consensus for disagreement. However, 46% of experts seem to express some confidence that neuroprotective treatment may be available in the future (Table 5).

In terms of investigational therapies, new formulations of oral levodopa were not considered useful in the treatment of PD nowadays or in the future, although they showed a trend to increase. New formulations of levodopa (intraduodenal, inhaled, subcutaneous, etc.) (86%), antidyskinetic drugs (79%), therapies for non-motor symptoms (dementia, depression, impulse control disorders [ICD], fatigue, pain, etc.) (79%), therapies to increase levodopa bioavailability (camicinal, DA-9701, VY-AADC01, etc.) (78%) and new symptomatic drugs (MAO-B, COMTI, agonists, etc.) (70%) were considered to be quite or very useful in the treatment of PD with a trend of increase by 2030 (Appendix A). When considering non-pharmacological therapies, the use of ultrasound (High-Intensity Focused Ultrasound [HIFU], Low-Intensity Focused Ultrasound [LIFU]) (88%) and DBS (72%) were considered quite or very useful in the treatment of PD and their utility was expected to increase by 2030. Other therapies (transcranial magnetic stimulation [TMS], transcranial direct current stimulation [tDCS], caloric vestibular stimulation, etc.) were considered to be of little or no use and their trend towards 2030 did not change. There was no consensus on the utility of surgical procedures other than DBS, either now or by 2030 (Appendix A). Other agents with a possible modifying effect on the progressive course of the disease such as immunotherapies and regenerative therapies, were considered not applicable nowadays, with no consensus on the trend towards 2030; however, the highest percentage of responses pointed out an increased use by 2030 (Appendix A).

### 3.8. Impact of PD

According to the experts, mental health, social life, quality of life and working life are affected in early PD patients, and more than 60% of the experts indicate that in 2030 the impact of these factors will remain unchanged. The same trend is observed among items considered to have low impact on PD patients in early stages (cognitive performance, autonomy in performing instrumental and basic activities of daily living and life expectancy) (Figure 2A).

In the advanced PD patient, experts considered that all items are affected by the disease on patients nowadays, and that this scenario will remain unchanged by 2030 (data not shown). Furthermore, the level of impact of PD on the caregiver was also high for all items considered, and more than 60% of experts indicated that this impact will remain unchanged (Figure 2B).

Regarding economic impact, the experts agreed that currently indirect costs (loss of productivity, home care and caregiver), second-line therapies and economic burdens have a high financial impact (Figure 3). Although experts did not reach a consensus, we observed a general trend towards remaining unchanged or an increase for all items.

### 3.9. Needs of Management

According to the experts, current needs regarding PD diagnosis include appropriate information from professionals, shortening the time to obtain a PD diagnosis, rapid referral to protocolized movement disorder units (MDU) and training primary care professionals for early detection of motor symptoms suggestive of PD. Satisfaction of all these items shows a trend to increase in the future, but experts only agreed that improvement will be seen in shortening the time to obtain PD diagnosis, and rapid referral to specific MDU (Table 6). Regarding statements of PD follow-up and treatment, all of them were considered needs nowadays, but no consensus in their future satisfaction was reached for any of the statements. Nevertheless, a trend of improvement in all items was observed (Table 6).

### 3.10. Resource Availability and General Needs

At present, there is limited availability of National Health System resources and Neurology Services in all the items consulted. With a view to 2030, there was no clear consensus on improvement, except for the availability of protocols and action plans, on which panelists agreed that these will increase in the future (80%) (Table 7).

Regarding the need for PD awareness, training, education, resources, economic impact and research, there was consensus on all items. Although it did not reach consensus for the future, panelists predicted that the satisfaction of all these needs will increase by 2030 (Table 7).

## 4. Discussion

The outcomes of this project provide data on current diagnosis and management of PD and its trends towards 2030 in Spain. This project included panelists with extensive experience in PD and the main results show a future perspective of PD management similar or better than nowadays.

PD is one of the most prevalent neurodegenerative diseases at present [1], and most experts (92%) considered that prevalence will increase by 2030, in accordance with estimates published in 2005, predicting that the number of individuals with PD would double by 2030 [4].

The Parkinson’s Disease Survey Observing the Quality of Care (Encuesta de Parkinson Observando la Calidad Asistencial, EPOCA) study carried out in Spain showed that more than half of patients get PD diagnosis between 1 and 5 years from the first symptom [17], which is quite delayed, and this gap significantly increases in patients with early-onset. Besides, according to SEN, up to 24% of PD patients have been misdiagnosed [3]. Therefore, efforts should be put towards an early and accurate diagnosis of PD, shortening the time to obtain PD diagnosis and rapid referral, as needs identified by study panelists.

Main criteria used for PD diagnosis are UK-PDSBB criteria and MDS criteria. When UK-PDSBB criteria are used, non-motor symptoms are not included, although these symptoms significantly contribute to health status and quality of life in PD [18]. In contrast, new MDS criteria [9] include motor symptoms and supportive criteria (non-motor symptoms). In our study, a trend towards a change to PD definition including non-motor or atypical symptoms, MDS criteria, and probably functional neuroimaging findings is observed.

Neuropsychiatric disorders (depression, apathy, sleep disturbances and anxiety) are common in PD patients; however, are often underrecognized and undertreated. In particular, >80% of PD patients develop dementia after 20 years, and hallucinations are consistently associated with progressive cognitive deterioration and dementia in PD [19,20,21]. According with data form the literature, our expert panel agreed that these symptoms would become relevant prognostic factors for PD patients in the future.

Given that PD is multifactorial and its etiopathogenesis is still unknown, no valid biomarkers for diagnosis are currently available. At the present time, 90 independent PD risk-associated mutations have been identified in more than 20 genes [22]; and it is still on-going research. Panelists also reported that geneticists will have a more relevant role in PD diagnosis, probably linked to the development of emerging technologies focused on the improvements of the knowledge of genetic risk, genetic predisposition and genetic features. Understanding the genetic factors that influence PD development will be important not only for PD diagnosis, but also for developing new and personalized treatments.

A range of biomarkers, including imaging, biochemical or genetic biomarkers have been proposed for PD diagnosis and prognosis; unfortunately, none can currently be used in real life [23]. Nevertheless, panelists rely on their potential for improving PD management, since they agreed that the use of CSF and plasma biomarkers and structural imaging as prognostic markers of the disease will increase by 2030. Biomarkers could also provide information on disease progression and treatment efficacy in the future, being involved in monitoring follow-up. A recent study has identified a CSF biomarker (α-synuclein) associated with increased risk of subsequent diagnosis of PD or dementia with Lewy bodies [10] approaching this promising future.

Apart from biomarkers, new sensor-based and wearable and non-wearable, hybrid technologies that can assess PD symptoms in an objective way will be useful for remote monitoring of PD patients in the future. Although nowadays its use is still limited and further research is needed, panelists agreed that they will become an important support in PD management [24,25].

The panelists determined that PD treatment in non-fluctuating patients should include more than one drug; however, up to 60% of the experts indicate that high-dose levodopa will continue to be used in monotherapy by 2030. Since its introduction in 1967, high-dose oral levodopa regimen for PD treatment remains uncontested [26].

In the fluctuating PD patients, treatment with high-dose levodopa will remain unchanged in the future, even though levodopa is associated with the appearance of motor complications. Since recent reports indicate that levodopa use does not modify the disease course, its use as antiparkinsonian drug is expected to continue to be maintained in the future, although low-dose levodopa is recommended to avoid dyskinesias. Current guidelines recommend levodopa in combination; these results suggest the need for better training about polytherapy benefits in these patients. There is also a trend towards a greater use of levodopa + safinamide and the combination of levodopa + DA+ safinamide not observed with other MAOI agents. This tendency might be explained by the proven effect of safinamide in reducing motor and non-motor fluctuations, its low rate of adverse events reported in several studies and importantly, the demonstrated effect of safinamide in alleviating non-motor symptoms, including pain and depression, and on quality of life impairment [27,28,29,30].

PD in later stages is characterized by the emergence of disabling motor and non-motor symptoms, which are poorly controlled with conventional therapies and have a significant impact on the patient’s health status [31]. In PD patients with an advanced stage, panelists observed a trend towards an increase in the use of new therapies, in particular DBS. This probably reflects that more patients will reach a device-based therapy phase, or that the available therapies will not modify the course of the disease. There is a significant need for developing treatments to modify the disease progression itself. New therapies under development such as stem cell therapies or gene-therapy [32] and a number of symptomatic and disease modifying therapies are currently being tested in clinical trials [33]. The application of high intense focused ultrasound (HIFU) is a quite new procedure that is being used for PD treatment. In our study, panelists agreed on its increased use in the future in spite of the fact that it is quite a new technique. They also considered HIFU as a useful tool for treating PD patients.

There is a general idea that both the appearance of motor and non-motor symptoms is relevant for deciding on a change of treatment; however, regarding symptoms beyond the CNS, experts did not consider them relevant when deciding on a change of treatment.

Guidelines state that a multidisciplinary team including professionals for assessing the personal, family, work and social situation of the patient and create personalized treatments for motor and non-motor symptoms with experience and adequate training is crucial for therapy success [34]. The group of experts considered that the role of the neurologist specialized in PD will become more relevant in the future, as well as the role of movement disorder specialist nurses, highlighting the importance of specialization.

As with many other neurologic conditions, PD has a great impact on quality of life and increases mortality among PD patients [35]. As the disability increases, PD symptoms and the disease course affect not only patients, but also their families and caregivers. Furthermore, it entails high economic burden with an estimated cost of €17,000 per patient per year in Spain [13]. Pharmacological treatment was identified as the main driver (34% of direct costs), and as the disease progresses, the need for symptomatic treatment grows.

PD unmet needs can be addressed through a comprehensive approach including physical, psychological, social and financial aspects. Early actions on diagnosis, follow-up and care planning can allow patients and caregivers to hamper the disease course and to develop anticipatory strategies. In this vein, a recent study identified a desire of PD patients and care partners to establish roadmaps as a guide for decision-making and planning [36].

### Limitations

The panelists participating in the study were representative of the clinical practice across Spain, and provided detailed information regarding PD clinical management. Consequently, results reflect the current management of PD and the vision for the future in a populous European country; however, the conclusions of this study cannot be extrapolated to other countries or cultural settings.

It has to be noted that the COVID-19 outbreak may have had an impact on the results, given that the project was truncated by this pandemic and carried out in two different scenarios. Besides, as with all Delphi studies, the results derived from the panel express an opinion and do not analyze prospective or retrospective data. Finally, the questionnaire was long, and bias due to tiredness cannot be excluded.

To our knowledge, this is the first study aiming to shed light on planning for future PD management in Spain. This study highlights the unmet needs in diagnosis and treatment and how crucial it is to establish recommendations on the diagnostic-therapeutic management of PD to optimize and design future management strategies.

## 5. Conclusions

The global and, in particular, socio-economic impact of PD will continue to be very important in all life dimensions of patients, family, caregivers, and in society in general, especially for patients in more advanced stages.

The experts considered that there will be an improvement in diagnostic capacity, follow-up and the development of innovative treatments, with fewer diagnostic errors and earlier diagnosis. This will be due to improved access of patients to movement disorder experts, and future studies on biomarkers will probably contribute to this improvement. In prognosis, the use of certain biomarkers will be important; in monitoring, new technologies will have an impact; and in treatment, depending on the evolutionary stage of the disease, the growing roles of expert nurses, neurologists, and other professionals (e.g., physiotherapists) will be very important, forming multidisciplinary teams that will move towards a more individualized and technological medicine. In summary, this work offers information that can help healthcare professionals to reflect individually on the care of PD in their area and propose future strategies to improve the management of patients and their environment (e.g., protocols, national plans, research management, social measures, etc.), as well as advance in knowledge and research.

## Figures and Tables

**Figure 1 brainsci-11-01027-f001:**
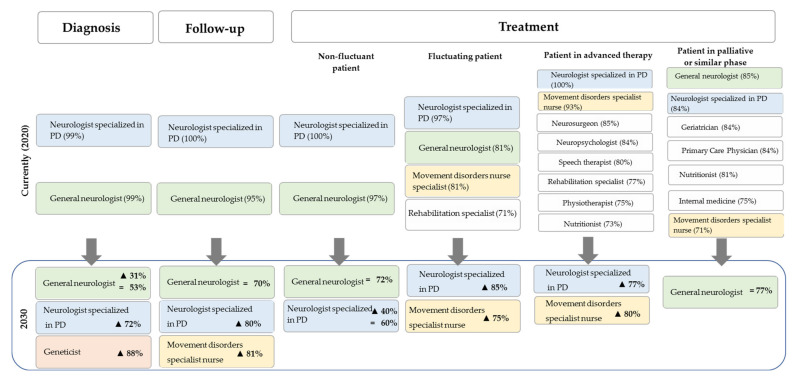
Relevance of medical specialties: present and future. For the current situation, percentages of panelists who considered a given specialty “Relevant” or “Very relevant” in each stage are shown. For the trend by 2030, percentages of panelists who predicted a decrease (▼), no change (=) or an increase (▲) for the relevance of each specialty in each stage are shown.

**Figure 2 brainsci-11-01027-f002:**
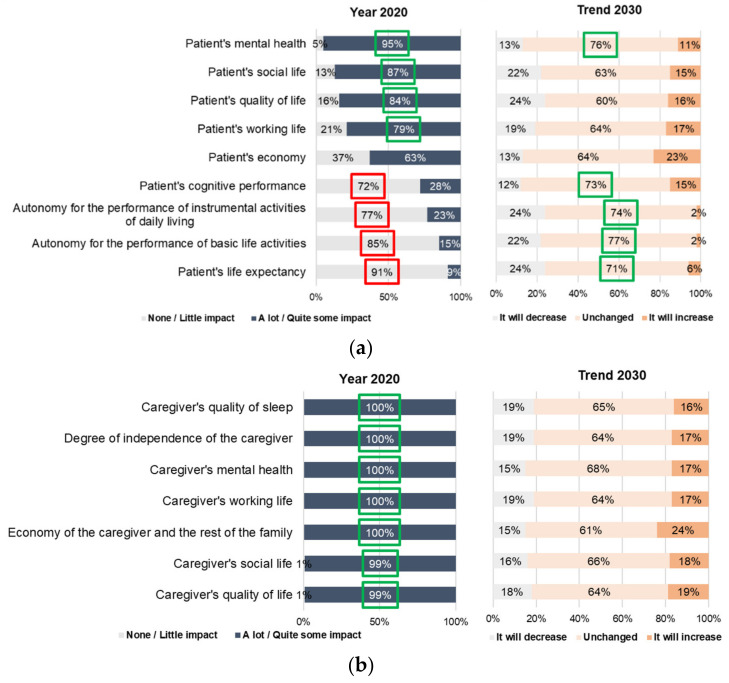
Impact of PD on patients and caregivers: (**a**) Impact of PD on patients (PD in early stages); (**b**) Impact of PD on caregivers.

**Figure 3 brainsci-11-01027-f003:**
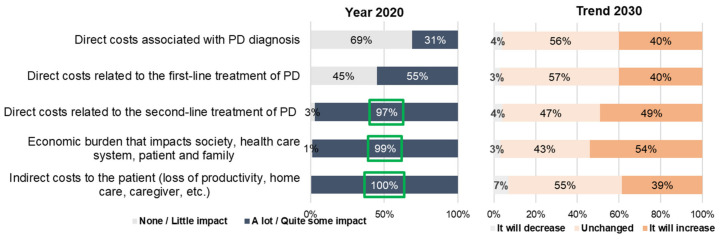
Economic impact of PD.

**Table 1 brainsci-11-01027-t001:** Prognostic factors: present and future (consensus after both rounds).

Prognostic Factors	YEAR 2020			TREND 2030			
	Currently, This Prognostic Factor for PD Is Used…			By 2030, the Use of This Prognostic Factor for PD Will…			
	Rarely/Never	Often/Always	Consensus	▼	=	▲	Consensus
Phenotype with axial involvement (blockings and balance problems)	0%	**100%**	Often/Always	0%	63%	37%	Undetermined
Cognitive impairment	1%	**99%**	Often/Always	0%	59%	41%	Undetermined
Early presence and severity of motor complications	1%	**99%**	Often/Always	1%	**73%**	26%	Not change
Age at onset	7%	**93%**	Often/Always	0%	**76%**	24%	Not change
Neuropsychiatric disorders (depression, apathy, hallucinations)	8%	**92%**	Often/Always	0%	48%	52%	Undetermined
Hoehn & Yahr clinical stage	15%	**85%**	Often/Always	19%	**77%**	5%	Not change
Autonomic alterations	47%	53%	Undetermined	3%	43%	55%	Undetermined
Sleep disorders (RBD associated to PD)	68%	32%	Undetermined	1%	60%	39%	Undetermined
CSF biomarkers	**97%**	3%	Rarely/Never	4%	19%	**77%**	Increase ▲
Serum molecular markers (inflammation, neurodegeneration, etc.)	**93%**	7%	Rarely/Never	1%	23%	**76%**	Increase ▲
Genetic alterations	**91%**	9%	Rarely/Never	1%	6%	**93%**	Increase ▲
Severe hyposmia or anosmia	**89%**	11%	Rarely/Never	0%	87%	13%	Not change
Degree of alteration in DATSCAN and PET studies, among others	**87%**	13%	Rarely/Never	6%	45%	49%	Undetermined
Use of structural imaging to determine the degree of alteration (e.g., atrophy in certain areas as predictors of dementia, etc.)	**84%**	16%	Rarely/Never	5%	17%	**78%**	Increase ▲

All panelists (*n* = 75) answered all questions. The percentage of panelists in each category and consensus after the second round of votes are shown. Bold % indicates that 70% consensus was achieved. CSF: Cerebrospinal fluid. DATSCAN: Dopamine Transporter Scan. PD: Parkinson’s disease. PET: Positron emission tomography. RBD: Rapid eye movement sleep behavior disorder or REM behavior disorder. **▼**: it will decrease; =: it will not change; **▲**: it will increase.

**Table 2 brainsci-11-01027-t002:** Criteria, scales and/or tests for the diagnosis of PD: present and future (consensus after both rounds).

Items	YEAR 2020			TREND 2030			
	Diagnosis of PD Is Currently Based on This Criterium, Marker, Parameter or Test.			By 2030, the Use of This Criterium, Marker, Parameter or Test for the Diagnosis of PD Will…			
	(Strongly) Disagree	(Strongly) Agree	Consensus	▼	=	▲	Consensus
Atypical signs	5%	**95%**	Agree	0%	**70%**	30%	Not change
New MDS criteria	17%	**83%**	Agree	5%	56%	39%	Undetermined
Non-motor symptoms (sense of smell…)	21%	**79%**	Agree	2%	36%	63%	Undetermined
Functional neuroimaging (SPECT, PET, heart scintigraphy)	21%	**79%**	Agree	0%	22%	**78%**	Increase ▲
Pharmacological tests (levodopa test, apomorphine test)	32%	68%	Undetermined	8%	**85%**	7%	Not change
Neurophysiological markers	**100%**	-	Disagree	13%	**72%**	15%	Not change
Hyposmia test	**99%**	1%	Disagree	15%	**73%**	12%	Not change
Biochemical markers	**97%**	3%	Disagree	3%	23%	**74%**	Increase ▲
Sleep analysis scales (PSG)	**95%**	5%	Disagree	10%	**73%**	17%	Not change
Neurosonography	**95%**	5%	Disagree	11%	49%	39%	Undetermined
Genetic criteria	**93%**	7%	Disagree	0%	30%	**70%**	Increase ▲
Structural neuroimaging	**93%**	7%	Disagree	4%	53%	43%	Undetermined

All panelists (*n* = 75) answered all questions. Percentages of panelists in each category and consensus after the second round of votes are shown. Bold % indicates that 70% consensus was achieved. MDS: Movement Disorder Society. PD: Parkinson’s disease. PET: Positron emission tomography. PSG: polysomnography. SPECT: Single-photon emission computed tomography. **▼**: it will decrease; =: it will not change; **▲**: it will increase.

**Table 3 brainsci-11-01027-t003:** Criteria, scales and/or tests for the follow-up of PD: present and future (percentages of respondents after both rounds).

Items	YEAR 2020			TREND 2030			
	Follow-Up of PD Is Currently Based on This Scale or Test.			By 2030, the use of This Scale or Test During Follow-Up of PD Will…			
	(Strongly) Disagree	(Strongly) Agree	Consensus	▼	=	▲	Consensus
Hoehn & Yahr scale	11%	**89%**	Agree	6%	**88%**	6%	Not change
MDS-UPDRS	19%	**81%**	Agree	5%	**77%**	18%	Not change
Cognitive impairment scales	20%	**80%**	Agree	0%	55%	45%	Undetermined
Wearing-off assessment scales	24%	**76%**	Agree	2%	67%	31%	Undetermined
Scales measuring impulse control behaviors	25%	**75%**	Agree	0%	57%	43%	Undetermined
Patient diary	27%	**73%**	Agree	5%	53%	42%	Undetermined
Dyskinesia rating scales	33%	67%	Undetermined	3%	**71%**	26%	Not change
Quality of life scales	37%	63%	Undetermined	0%	56%	44%	Undetermined
Motor symptoms scales	39%	61%	Undetermined	0%	69%	31%	Undetermined
Non-motor symptoms scales	43%	57%	Undetermined	0%	39%	61%	Undetermined
Disability scales	45%	55%	Undetermined	0%	60%	40%	Undetermined
Neuropsychiatric disorders scales	49%	51%	Undetermined	0%	52%	48%	Undetermined
Pharmacogenetics	**100%**	-	Disagree	3%	25%	**72%**	Increase ▲
Biochemical markers	**99%**	1%	Disagree	1%	28%	**71%**	Increase ▲
Inflammation markers	**97%**	3%	Disagree	3%	42%	55%	Undetermined
Neurosonography	**96%**	4%	Disagree	10%	57%	33%	Undetermined
Structural neuroimaging	**95%**	5%	Disagree	6%	52%	42%	Undetermined
Wearable, non-wearable, hybrid and smartphone sensors	**93%**	7%	Disagree	0%	6%	**94%**	Increase ▲
Functional neuroimaging	**91%**	9%	Disagree	1%	37%	62%	Undetermined
Pharmacological tests (levodopa test, apomorphine test)	**88%**	12%	Disagree	17%	**80%**	3%	Not change
Sleep analysis scales	**87%**	13%	Disagree	0%	60%	40%	Undetermined
Gait assessment scales	**76%**	24%	Disagree	2%	60%	38%	Undetermined

All panelists (*n* = 75) answered all questions. Percentages of panelists in each category and consensus after the second round of votes are shown. Bold % indicates that 70% consensus was achieved. MDS-UPDRS: Movement Disorder Society—Unified Parkinson’s Disease Rating Scale. **▼**: it will decrease; =: it will not change; **▲**: it will increase.

**Table 4 brainsci-11-01027-t004:** Treatment by patient profile: future trends.

TREATMENT	TREND 2030			
By 2030, the Use of This Treatment for Non-Fluctuating Patients Will…	▼	=	▲	Consensus
Non-pharmacological treatment alone ^a^	10%	**76%**	14%	Not change
**Monotherapy**				
MAOI ^b^	13%	**88%**	0%	Not change
DA ^c^	22%	**76%**	1%	Not change
Levodopa (<600 mg)	14%	**75%**	11%	Not change
Anticholinergics	35%	65%	0%	Undetermined
Levodopa (>600 mg)	35%	61%	4%	Undetermined
MAOI ^d^	1%	43%	56%	Undetermined
**Two-Drug Combination**				
Levodopa + MAOI ^b^	10%	**85%**	6%	Not change
MAOI ^b^ + DA	17%	**78%**	6%	Not change
Levodopa + DA ^c^	21%	**75%**	4%	Not change
Levodopa + COMTI	11%	**70%**	19%	Not change
MAOI ^d^ +DA	3%	46%	51%	Undetermined
Levodopa + MAOI ^d^	0%	38%	63%	Undetermined
**Three-Drug Combination**				
MAOI ^b^ + DA + levodopa	15%	**83%**	1%	Not change
COMTI +DA + levodopa	11%	**77%**	11%	Not change
MAOI ^d^ +DA + levodopa	6%	48%	46%	Undetermined
Polytherapy				
≥4 drugs	8%	**80%**	11%	Not change
**By 2030, the Use of This Treatment for Fluctuating Patients Will…**	**▼**	**=**	**▲**	**Consensus**
**Monotherapy**				
Levodopa (<600 mg)	27%	72%	1%	Not change
AD ^c^	30%	69%	1%	Undetermined
Levodopa (>600 mg)	34%	63%	3%	Undetermined
**Two-Drug Combination**				
Levodopa + AD	10%	**83%**	7%	Not change
Levodopa + COMTI	1%	**79%**	20%	Not change
Levodopa + MAOI ^b^	23%	**75%**	3%	Not change
Levodopa + MAOI ^d^	1%	35%	63%	Undetermined
**Three-Drug Combination**				
MAOI ^b^ +DA + levodopa	13%	**79%**	8%	Not change
COMTI +DA + levodopa	1%	**79%**	20%	Not change
MAOI ^d^ +DA + levodopa	0%	41%	59%	Undetermined
**Polytherapy**				
≥4 drugs	3%	77%	20%	Not change
**By 2030, the Use of This Treatment in Advanced Therapy Will…**	**▼**	**=**	**▲**	**Consensus**
DBS + other pharmacological treatments	3%	21%	**76%**	Increase ▲
Carbidopa/levodopa (enteral) + other pharmacological treatments	7%	25%	68%	Undetermined
Apomorphine s.c. + other pharmacological treatments	3%	45%	52%	Undetermined

All panelists (*n* = 75) answered all questions. Percentages of panelists in each category and consensus after the second round of votes are shown. Bold % indicates that 70% consensus was achieved. COMTI: catechol-O-methyltransferase inhibitors. DA: dopaminergic agonists. DBS: deep brain stimulation. MAOI: monoaminooxidase inhibitors. s.c.: subcutaneous. ^a^ Physiotherapy, speech therapy, rehabilitation, etc. ^b^ Rasagiline, Selegiline. ^c^ Rotigotine, Ropinirole, Pramipexole. ^d^ Safinamide. **▼**: it will decrease; =: it will not change; **▲**: it will increase.

**Table 5 brainsci-11-01027-t005:** Change/modification of PD treatment: present and future.

Items	YEAR 2020			TREND 2030			
	Onset of These Symptoms and/or Comorbidities Is a Reason for Changing PD Treatment.			By 2030, the Relevance of These Symptoms and/or Comorbidities as a Reason to Change Treatment Will…			
Symptoms and/or Comorbidities	(Strongly) Disagree	(Strongly) Agree	Consensus	▼	=	▲	Consensus
Neuropsychiatric and cognitive disorders ^a^	-	**100%**	Agree	0%	55%	45%	Undetermined
Deterioration in BADL autonomy	-	**100%**	Agree	0%	60%	40%	Undetermined
Motor symptoms ^b^	-	**100%**	Agree	1%	63%	36%	Undetermined
Impaired quality of life	1%	**99%**	Agree	0%	54%	46%	Undetermined
Affective, emotional and volitional disorders ^c^	1%	**99%**	Agree	0%	59%	41%	Undetermined
Sleep disorders ^d^	5%	**95%**	Agree	0%	62%	38%	Undetermined
Autonomic disorders ^e^	9%	**91%**	Agree	0%	54%	46%	Undetermined
Disorders of other organs outside the CNS ^f^	84%	16%	Disagree	2%	83%	15%	Not change
**Statements Regarding Change/Modification of PD Treatment**							
	**(Strongly) Disagree**	**(Strongly) Agree**	**Consensus**	**▼**	**=**	**▲**	**Consensus**
Treatment is started at early stages	-	**100%**	Agree	0%	49%	51%	Undetermined
Pharmacological treatment is modified when an impairment of quality of life is observed.	-	**100%**	Agree	0%	57%	43%	Undetermined
Treatment is changed when cognitive dysfunction and/or psychotic symptoms appear.	1%	**99%**	Agree	0%	68%	32%	Undetermined
Pharmacological treatment is changed when new motor symptoms appear.	1%	**99%**	Agree	0%	**73%**	27%	Not change
Treatment is modified if lack of response to a new treatment is observed.	1%	**99%**	Agree	0%	**74%**	26%	Not change
Treatment is changed in case of side effects.	3%	**97%**	Agree	0%	**71%**	29%	Not change
Pharmacological treatment is changed when new non-motor symptoms appear.	5%	**95%**	Agree	0%	49%	51%	Undetermined
Neuroprotective treatments prevail over symptomatic treatments	**96%**	4%	Disagree	3%	51%	46%	Undetermined

All panelists (*n* = 75) answered all questions. Percentages of panelists in each category and consensus after the second round of votes are shown. Bold % indicates that 70% consensus was achieved. BADL: basic activities of daily living. CNS: Central nervous system. PD: Parkinson’s disease. **▼**: it will decrease; =: it will not change; **▲**: it will increase. ^a^ Hallucinations, delusions, delusional ideation, dementia, difficulties in concentration, etc. ^b^ Motor fluctuations, bradykinesia, freezing of gait, balance alterations, postural reflex alterations, dysphagia, hypersalivation, dysarthria, rigidity, etc. ^c^ Depression, apathy, anxiety, sexual disorders, etc. ^d^ REM behavior disorder, insomnia, excessive daytime sleepiness, etc. ^e^ Constipation, salivation, hypotension, incontinence, erectile dysfunction, sweating, sexual disorders, swallowing disorders, etc. ^f^ Osteoporosis, back pain, respiratory disorders, etc.

**Table 6 brainsci-11-01027-t006:** Needs of PD management: present and future.

**Items**	**YEAR 2020**			**TREND 2030**			
	**This Item Is Important in Current Diagnosis of PD**			**By 2030, the Importance of This Item on Diagnosis of PD Will…**			
**PD Diagnosis**	**Not (very) important**	**(Very) important**	**Consensus**	**▼**	**=**	**▲**	**Consensus**
Appropriate information from professionals	1%	**99%**	Important	0%	37%	63%	Undetermined
Shortening the time to obtain PD diagnosis	9%	**91%**	Important	1%	19%	**79%**	Increase ▲
Rapid referral to protocolized MDU	15%	**85%**	Important	0%	22%	**78%**	Increase ▲
Training primary care professionals for early detection of motor symptoms suggestive of PD	9%	**81%**	Important	0%	36%	64%	Undetermined
Providing psychological support to the patient, family and/or caregiver	37%	63%	Undetermined	0%	39%	61%	Undetermined
Support from movement disorders specialist nurses	59%	41%	Undetermined	1%	17%	**81%**	Increase ▲
Early diagnosis based on non-motor symptoms	69%	31%	Undetermined	0%	23%	**77%**	Increase ▲
**PD Follow-Up**							
	**This Item Is Needed during the Follow-Up of PD**			**By 2030 the Level of Satisfaction of This Need Will…**			
	**(Strongly) Disagree**	**(Strongly) Agree**	**Consensus**	**▼**	**=**	**▲**	**Consensus**
Ensuringthe resolution of any doubts the patient may have.	3%	**97%**	Agree	0%	45%	55%	Undetermined
Developing improved coordination between health professionals and social services	5%	**95%**	Agree	0%	35%	65%	Undetermined
Developing greater coordination between the different levels of health services	8%	**92%**	Agree	0%	33%	67%	Undetermined
Improve the patient’s adaptation to the PD evolution	8%	**92%**	Agree	0%	46%	54%	Undetermined
Establishing a follow-up protocol in the first months after diagnosis.	15%	**85%**	Agree	2%	51%	47%	Undetermined
Providing legal and medical assistance	15%	**85%**	Agree	0%	50%	50%	Undetermined
**PD Treatment**							
	**This Item Is Needed for Treatment of PD**			**By 2030, the Level of Satisfaction of This Need Will…**			
	**(Strongly) Disagree**	**(Strongly) Agree**	**Consensus**	**▼**	**=**	**▲**	**Consensus**
Availability of speech therapy sessions to help with language and information processing	1%	**99%**	Agree	0%	41%	59%	Undetermined
Availability of dopaminergic treatment that provides the benefits of levodopa without worsening motor complications	3%	**97%**	Agree	1%	34%	64%	Undetermined
Availability of occupational therapy that provides adaptive strategies	4%	**96%**	Agree	0%	44%	56%	Undetermined
Availability of global and integrated rehabilitative treatment that allows less medication intake (number and dose)	5%	**95%**	Agree	0%	32%	68%	Undetermined
Availability of physiotherapy and other alternative therapies such as hydrotherapy, yoga, pilates or Tai-Chi	5%	**95%**	Agree	0%	35%	65%	Undetermined
Having an effective surgical treatment for PD in advanced stages to rescue the patient	8%	**92%**	Agree	1%	43%	55%	Undetermined
Early initiation of appropriate treatment to delay PD progression and improve quality of life	11%	**89%**	Agree	0%	46%	54%	Undetermined

All panelists (*n* = 75) answered all questions. Percentages of panelists in each category and consensus after the second round of votes are shown. Bold % indicates that 70% consensus was achieved. MDU: movement disorders unit. PD: Parkinson’s disease. **▼**: it will decrease; =: it will not change; **▲**: it will increase.

**Table 7 brainsci-11-01027-t007:** Resource availability and general needs: present and future.

**Items**	**YEAR 2020**			**TREND 2030**			
	**Currently There Is Sufficient Availability of This Resource.**			**By 2030, Availability of This Resource Will…**			
**Resources in NHS and Neurology Services**	**(Strongly Disagree**	**(Strongly) Agree**	**Consensus**	**▼**	**=**	**▲**	**Consensus**
Number of nurses specialized in movement disorders	**99%**	1%	Disagree	4%	31%	65%	Undetermined
New technologies available	**97%**	3%	Disagree	1%	33%	66%	Undetermined
Number of daycare centers for patients with PD	**97%**	3%	Disagree	5%	36%	59%	Undetermined
Time dedicated to the medical visit with the patient	**97%**	3%	Disagree	1%	59%	40%	Undetermined
Health care protocols and action plans	**95%**	5%	Disagree	0%	20%	**80%**	Increase ▲
Number of MDU units	**95%**	5%	Disagree	0%	38%	62%	Undetermined
Number of neurologists specialized in PD	**95%**	5%	Disagree	1%	44%	55%	Undetermined
**General Needs**							
	**This Item Is a General Need in Management of PD**			**By 2030 the Level of Satisfaction of This Need Will…**			
	**(Strongly) Disagree**	**(Strongly) Agree**	**Consensus**	**▼**	**=**	**▲**	**Consensus**
**PD Awareness**							
Make patients visible to raise awareness and knowledge of PD	5%	**95%**	Agree	0%	39%	61%	Undetermined
Work together on policies for the patient’s reintegration into society, both socially and in the workplace	8%	**92%**	Agree	0%	42%	58%	Undetermined
Raise awareness of the notable role that patient associations have in the treatment of PD and collaborate with them (e.g., raising public awareness).	12%	**88%**	Agree	0%	35%	65%	Undetermined
**Training and Education**							
Adequately train caregivers in the management of the patient and the disease	1%	**99%**	Agree	0%	36%	64%	Undetermined
Adequately train patients in the management of their own disease	1%	**99%**	Agree	0%	38%	62%	Undetermined
Promote and provide good training for neurologists who assess the degree of disability	7%	**93%**	Agree	0%	49%	51%	Undetermined
Promote and provide good training for PCPs who assess the degree of disability	11%	**89%**	Agree	1%	55%	43%	Undetermined
**Resources**							
Have specialized protocols aimed at PCP and general neurologists	4%	**96%**	Agree	0%	46%	54%	Undetermined
Ensure proper counseling and psychological support for the patient and family throughout the disease course	4%	**96%**	Agree	0%	49%	51%	Undetermined
Optimize the ratio patients per specialist (neurologists…)	4%	**96%**	Agree	1%	51%	47%	Undetermined
Provide more resources to hospital centers (MDU, etc.)	5%	**95%**	Agree	1%	45%	54%	Undetermined
Involve more medical specialties (primary care, geriatrics, etc.) in patient management	8%	**92%**	Agree	0%	59%	41%	Undetermined
**Economic Impact**							
Optimize existing resources while minimizing the economic impact on society and on the health system	3%	**97%**	Agree	3%	56%	41%	Undetermined
Reduce the economic impact on the caregiver	3%	**97%**	Agree	5%	58%	37%	Undetermined
Reduce the economic impact on the patient	3%	**97%**	Agree	5%	59%	36%	Undetermined
**Research**							
Provide financial support for the research of new treatments for PD	4%	**96%**	Agree	3%	31%	67%	Undetermined
Carry out detailed cost studies, including some intangible costs of notorious importance (e.g., depression) as well as the costs associated with each stage of the disease	9%	**91%**	Agree	1%	44%	54%	Undetermined

All panelists (*n* = 75) answered all questions. Percentages of panelists in each category and consensus after the second round of votes are shown. Bold % indicates that 70% consensus was achieved. MDU: movement disorder units. PCP: primary care physician. PD: Parkinson’s disease. **▼**: it will decrease; =: it will not change; **▲**: it will increase.

## Data Availability

The data presented in this study are available on request from the corresponding author. The data are not publicly available due to privacy.

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
