# Peer review of "Present and Future of Parkinson’s Disease in Spain: PARKINSON-2030 Delphi Project"

_brainsci, 2021, doi:10.3390/brainsci11081027_

Round 1
Reviewer 1 Report
Thanks for recommending me as a reviewer. The authors were used a modified Delphi method to better understand the management and the socioeconomic burden of PD in Spain. The methods and results of the study are interesting. If the authors complete minor revisions, the quality of the study will be further improved.
- The introduction section is well written. If the author writes more specifically about the need for Delphi research in the introduction section, it can help the reader understand.
2. line 118-123: Authors should be more specific about the "modified Delphi method" in the Methods section. What is the difference with "Delphi method"?
3. Authors should include suggestions (or implications) for subsequent researchers in the Conclusion section.
Author Response
Reviewer #1 (Comments to the Author):
Thanks for recommending me as a reviewer. The authors were used a modified Delphi method to better understand the management and the socioeconomic burden of PD in Spain. The methods and results of the study are interesting. If the authors complete minor revisions, the quality of the study will be further improved.
- The introduction section is well written. If the author writes more specifically about the need for Delphi research in the introduction section, it can help the reader understand.
AUTHORS – Thank you very much for your comment. We clarify what is Delphi method: “Considering the high incidence of the disease, its increased prevalence due to population ageing, and the unmet needs regarding diagnosis and treatment, it is crucial to better understand PD management to optimize and design future strategies. Thus, the aim of the PARKINSON 2030 project was to discuss and reach consensus using Delphi method (a process used to arrive at a group opinion or decision by surveying a panel of experts in which the experts respond to several rounds of questionnaires, and the responses are aggregated and shared with the group after each round) among a panel of experts in movement disorders on the current situation and their 10-year forecast of the general management of PD in daily clinical practice and to establish recommendations on the diagnostic and therapeutic management of PD”.
- line 118-123: Authors should be more specific about the "modified Delphi method" in the Methods section. What is the difference with "Delphi method"?
AUTHORS – Thank you very much for your comment. We clarify this question: “We used specifically a modified Delphi method (a group consensus strategy that systematically uses literature review, opinion of stakeholders and the judgment of experts within a field to reach agreement) [14–16] to better understand the management and the socioeconomic burden of PD in Spain”.
- Authors should include suggestions (or implications) for subsequent researchers in the Conclusion section.
AUTHORS – Thank you very much for your comment. After all suggestions already described, we added the final sentence: “In summary, this work offers information that can help healthcare professionals to reflect individually on the care of PD in their area and propose future strategies to improve the management of patients and their environment (e. g., protocols, national plans, research management, social measures, etc.), as well as advance in the knowledge and research”.

Reviewer 2 Report
This article reviews an impressive project of Consensus building through Modified Delphi method for future directions of care and burden of Parkinson Disease in Spain.
The project involved an impressive amount of 75 Movement Disorders experts nationally and 7 meetings were concluded (instead of planned 12 due to Covid but still an impressive number of meetings).
Data is nicely summarized in extensive tables. A consensus cutoff was defined and is used to define results but individual percentages are also shared for further details which is very useful. Impact of PD and economic impact is also very importantly included in the discussions and summarized.
One can disagree or challenge the results on some of the items but the value of a consensus answer from multiple experts cannot be denied. Many useful ideas and concepts are reviewed by the experts for consensus making and can guide further research.
Author Response
Reviewer #2 (Comments to the Author):
This article reviews an impressive project of Consensus building through Modified Delphi method for future directions of care and burden of Parkinson Disease in Spain.
The project involved an impressive amount of 75 Movement Disorders experts nationally and 7 meetings were concluded (instead of planned 12 due to Covid but still an impressive number of meetings).
Data is nicely summarized in extensive tables. A consensus cutoff was defined and is used to define results but individual percentages are also shared for further details which is very useful. Impact of PD and economic impact is also very importantly included in the discussions and summarized.
One can disagree or challenge the results on some of the items but the value of a consensus answer from multiple experts cannot be denied. Many useful ideas and concepts are reviewed by the experts for consensus making and can guide further research.
AUTHORS – Thank you very much for your comment. We agree with you that this work took a significant effort and time and that the results are of interest. Many thanks again.
